# Vocal biomarker predicts fatigue in people with COVID-19: results from the prospective Predi-COVID cohort study

Abir Elbéji [ID],[1] Lu Zhang,[1] Eduardo Higa,[1] Aurélie Fischer,[1] Vladimir Despotovic,[2] Petr V Nazarov,[2] Gloria Aguayo [ID],[1] Guy Fagherazzi [ID][1]

[1]Department of Precision Health, Luxembourg Institute of Health, 1A-B, rue Thomas Edison, L-1445 Strassen, Luxembourg
[2]Bioinformatics Platform, Luxembourg Institute of Health, 1A-B, rue Thomas Edison, L-1445 Strassen, Luxembourg

**Correspondence to**
Dr Guy Fagherazzi;
guy.fagherazzi@lih.lu

## ABSTRACT

**Objective** To develop a vocal biomarker for fatigue monitoring in people with COVID-19.
**Design** Prospective cohort study.
**Setting** Predi-COVID data between May 2020 and May 2021.
**Participants** A total of 1772 voice recordings were used to train an AI-based algorithm to predict fatigue, stratified by gender and smartphone's operating system (Android/iOS). The recordings were collected from 296 participants tracked for 2 weeks following SARS-CoV-2 infection.
**Primary and secondary outcome measures** Four machine learning algorithms (logistic regression, k-nearest neighbours, support vector machine and soft voting classifier) were used to train and derive the fatigue vocal biomarker. The models were evaluated based on the following metrics: area under the curve (AUC), accuracy, F1-score, precision and recall. The Brier score was also used to evaluate the models' calibrations.
**Results** The final study population included 56% of women and had a mean (±SD) age of 40 (±13) years. Women were more likely to report fatigue (p<0.001). We developed four models for Android female, Android male, iOS female and iOS male users with a weighted AUC of 86%, 82%, 79%, 85% and a mean Brier Score of 0.15, 0.12, 0.17, 0.12, respectively. The vocal biomarker derived from the prediction models successfully discriminated COVID-19 participants with and without fatigue.
**Conclusions** This study demonstrates the feasibility of identifying and remotely monitoring fatigue thanks to voice. Vocal biomarkers, digitally integrated into telemedicine technologies, are expected to improve the monitoring of people with COVID-19 or Long-COVID.
**Trial registration number** NCT04380987.

## STRENGTHS AND LIMITATIONS OF THIS STUDY

⇒ This is the first study supporting the hypothesis that fatigue can be accurately monitored based on voice in people with COVID-19.
⇒ The analyses were based on a multilingual database of standardised voice recordings collected in real life from people with confirmed SARS-CoV-2 infection as determined by PCR.
⇒ There is no similar dataset available yet in the literature to replicate our findings.
⇒ The vocal biomarker is trained on a binary outcome (fatigue, yes/no) and does not reflect the entire spectrum of fatigue severity. Further work should be performed in that direction.

focus on the more important and urgent patients, was, and still is, critical.

This outbreak continues to impact people, with many patients suffering from a range of acute symptoms, such as fatigue. Fatigue is a common symptom in patients with COVID-19 that can impact their quality of life, treatment adherence and can be associated with numerous complications.[3] Recent findings showed that fatigue is a major symptom of the frequently reported Long-COVID syndrome. After recovering from the acute disease caused by the SARS outbreak, up to 60% of patients reported chronic fatigue 12 months later.[4] This supports the need for long-term monitoring solutions for these patients.

In general, fatigue can be of two types: physical and mental[5] experiencing lack of energy, inability to start and perform everyday activities and lack of desire to do things. In the context of COVID-19, determinants of fatigue were categorised as both central and psychological factors, the latest might also be indirectly caused by pandemic-related fear and anxiety.[6 7]

Fatigue affects men and women differently and has previously been shown to be reported differently in the two genders. Men and women have different anatomy

## INTRODUCTION

COVID-19 is a global outbreak. More than 199 million confirmed cases of COVID-19 have been detected worldwide as of 4 August 2021, with more than 4 million deaths reported by the WHO.[1] The worldwide population and healthcare systems have been greatly impacted by the COVID-19 pandemic. The pandemic has essentially put whole healthcare systems under pressure, requiring national or regional lockdowns.[2] Finding solutions that allow healthcare providers to

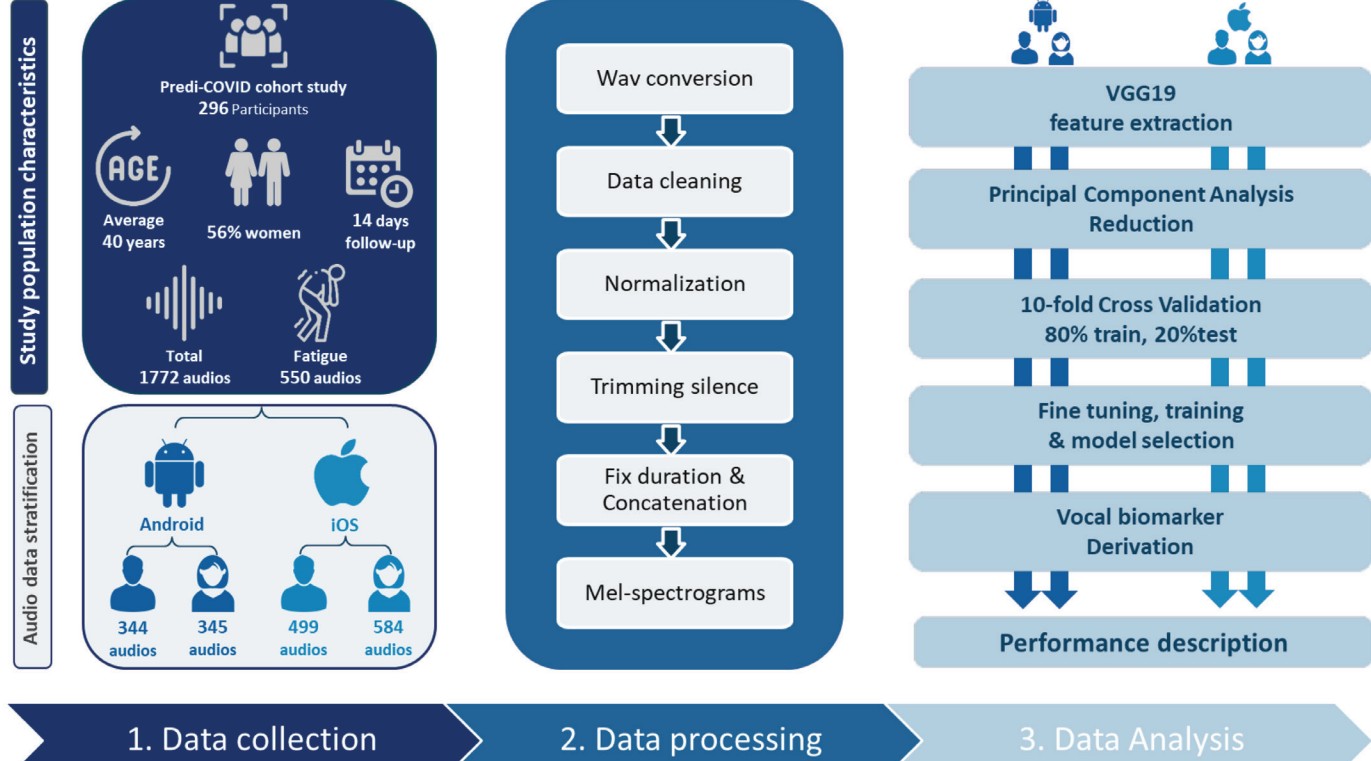

**Figure 1** General pipeline.

and physiology, resulting in significant sex differences in fatigability.[8]

Telemedicine, artificial intelligence (AI) and big data predictive analytics are examples of digital health technologies that have the potential to minimise the damaging effects of COVID-19 by improving responses to public health problems at a population level.[9]

Using telemonitoring technologies to enable self-surveillance and remote monitoring of symptoms might therefore help to improve and personalise COVID-19 care delivery.[10]

Voice is a promising source of digital data since it is rich, user-friendly, inexpensive to collect and non-invasive, and can be used to develop vocal biomarkers that characterise disease states. Previous research was mostly conducted in the field of neurodegenerative diseases, such as Parkinson's disease[11] and Alzheimer's disease.[12] There are also studies that confirm the relation of voice disorders to fatigue, for example, in chronic fatigue syndrome (CFS). Neuromuscular, neuropsychological and hormonal dysfunction associated with CFS can influence the phonation and articulation, and alter tension, viscosity and thickness of the tissue of the larynx, tongue and lips, leading to decreased voice quality.[13] Increased fatigue affects voice characteristics, such as pitch, word duration[14] and timing of articulated sounds.[15] Vocal changes related to fatigue are more observed in consonant sounds that require a high average airflow.[16]

In the context of the COVID-19 pandemic, respiratory sounds (eg, coughs, breathing and voice) are also used as sources of information to develop COVID-19 screening tools.[17–19] However, no previous work has been devoted to investigating the association of voice with COVID-19 symptoms.

We hypothesised that there is an association between fatigue and voice in patients with COVID-19 and that it is possible to train an AI-based model to identify fatigue and subsequently generate a digital vocal biomarker for fatigue monitoring. We used data from the large hybrid prospective Predi-COVID cohort study to investigate this hypothesis.

## METHODS

### Study design

This project uses data from the Predi-COVID study.[20] Predi-COVID is a hybrid cohort study that started in May 2020 in Luxembourg and involved participants who should meet all of the following requirements: (1) a signed informed consent form; (2) participants with confirmed SARS-CoV-2 infection as determined by PCR at one of Luxembourg's certified laboratories and (3) 18 years and older.

This study combines data from the national surveillance system, which is used for virtually all COVID-19 positive patients. Biological sampling, electronic patient-reported outcomes and smartphone voice recording were collected to identify vocal biomarkers of respiratory syndromes and fatigue in this study. More details about the Predi-COVID study can be found elsewhere.[20]

Health Inspection collaborators made the initial phone contact with potential participants. Those who consented

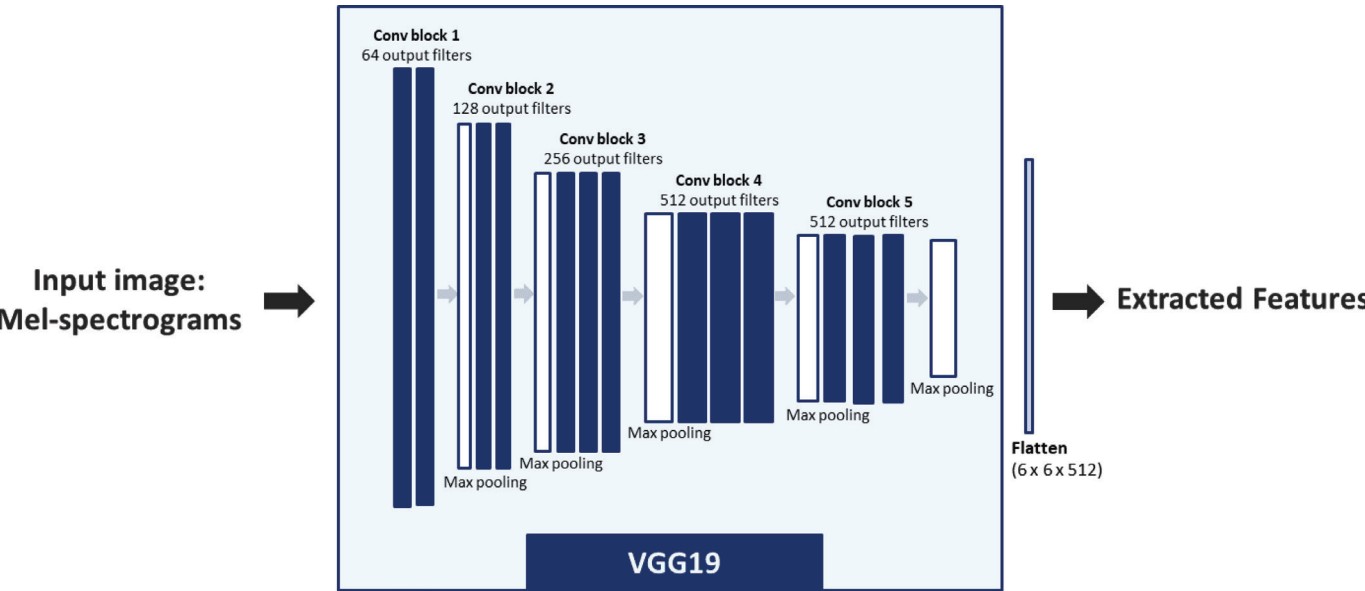

**Figure 2** VGG19 feature extraction.

to participate were contacted by a qualified nurse from the Clinical and Epidemiological Investigation Centre (Luxembourg Institute of Health), who outlined the study and arranged home or hospital visits.

### Patient and public involvement

The Predi-COVID initiative was an emergency response from national research institutions grouped under 'Research Luxembourg' to fight the COVID-19 pandemic in Luxembourg and contribute to the general effort in the crisis. Therefore, for timing and safety reasons, patients with COVID-19 were not directly included to participate in the study design. However, the first participants included in Predi-COVID provided feedback on general workflow, data collection, questionnaires and sampling, which was taken into account in an amendment to the protocol.[20]

### Data collection

Participants were followed for up to a year using a smartphone app to collect voice data. To ensure a minimum quality level, participants were asked to record it in a quiet environment while maintaining a certain distance from the microphone, and an audio example of what was required was also provided.

All the participants of this study were invited to record two audio types. The first, type 1 audio, required participants to read paragraph 1 of article 25 of the Declaration of Human Rights,[21] in their preferred language: French, German, English or Portuguese; and the second, type 2 audio, required them to hold the (a) vowel phonation without breathing for as long as they could (see online supplemental material 1 for more details).

Predi-COVID collects data in conformity with the German Society of Epidemiology's best practices guidelines.[22] To draft the manuscript, we followed the TRIPOD criteria (the Transparent Reporting of a multivariable prediction model of Individual Prognosis Or Diagnosis) for reporting AI-based model development and validation, as well as the corresponding checklist.

All Predi-COVID participants recruited between May 2020 and May 2021 who reported their fatigue status ("I feel well" as "No Fatigue" and "I am fatigued"/"I don't feel well" as "Fatigue") on the same day as the audio recordings during the 14 days of follow-up were included in this study.[23] As a result, several audio recordings for a single participant were available for both audio types.[24]

### Audio characteristics and vocal biomarker training

The audio recordings were collected in two formats, 3gp format (Android devices) and m4a format (iOS devices). Based on the smartphone's operating system and the user's gender (male/female), we trained one model for each category. This stratification was performed to minimise data heterogeneity and deal with sex as a potential confounding bias.

### Audio preprocessing

All of the raw audio recordings were preprocessed (figure 1). They were initially converted to .wav files, with audios lasting less than 2 s being excluded. Then, an audio clustering (DBSCAN) on basic features (duration, average, sum and SD of signal power, and fundamental frequency) was performed to detect outliers that were manually checked while excluding poor quality audios with (1) too noisy, (2) incorrect text reading, (3) type 1 and type 2 audios mixed or (4) extended silence in the middle. Finally, peak normalisation was used to boost the volume of quiet audio segments, and leading and trailing silences longer than 350 ms were trimmed.

### Feature extraction

We used transfer learning for the feature extraction process since it is adapted for small training databases.[25] Transfer

**Table 1** Study population characteristics the clinical data in the table above describe the overall population of the study

| | | m4a | | 3gp | | |
|---|---|---|---|---|---|---|
| | **All** | **Female** | **Male** | **Female** | **Male** | **P value (m4a, 3gp)** |
| Participants (N) | | | | | | |
| Total | 296 | 107 | 80 | 51 | 58 | – |
| Age (years) | | | | | | |
| Mean (SD) | 40.3 (12.6) | 38.8 (13.4) | 42.9 (12.7) | 37.8 (11.6) | 41.5 (11.3) | 0.28 |
| Body mass index (kg/m²) | | | | | | |
| Mean (SD) | 24.1 (4.7) | 24.6 (5.5) | 26.5 (4.1) | 24.1 (3.8) | 26.6 (4.17) | 0.95 |
| Antibiotic (%) | | | | | | |
| No | 265 (90) | 93 (87) | 73 (91) | 44 (86) | 55 (95) | 0.87 |
| Yes | 31 (10) | 14 (13) | 7 (9) | 7 (14) | 3 (5) | |
| Asthma (%) | | | | | | |
| No | 284 (96) | 104 (97) | 75 (94) | 47 (92) | 58 (100) | 0.82 |
| Yes | 12 (4) | 3 (3) | 5 (6) | 4 (8) | 0 (0) | |
| Smoking (%) | | | | | | |
| Never | 199 (67) | 77 (72) | 51 (64) | 36 (71) | 35 (60) | 0.41 |
| Former smoker | 53 (18) | 19 (18) | 20 (25) | 9 (18) | 13 (22) | |
| Current smoker | 44 (15) | 11 (10) | 9 (11) | 6 (11) | 10 (18) | |
| Audio recordings | | | | | | |
| Total | 1772 | 584 | 499 | 345 | 344 | <0.001 |
| No Fatigue | 1222 (69) | 394 (67) | 370 (74) | 190 (55) | 268 (78) | |
| Fatigue | 550 (31) | 190 (33) | 129 (26) | 155 (45) | 76 (22) | |
| Mean (SD) and maximum of audio recording per participant in the 14-day follow-up period | | | | | | |
| Mean (SD) | 6 (5) | 6 (5) | 6 (5) | 6 (5) | 6 (5) | – |
| Max | 16 | 14 | 16 | 15 | 14 | |

The total number and its percentage are used to represent all categorical data. The table summarises general information for describing audio data. All p values comparing iOS (m4a) and Android users (3gp) were calculated using $\chi^2$ test and Student's t-test.

learning is a technique where a model is constructed and trained with a set containing a large amount of data and then transfer and apply this learning to our dataset on top of it. It has the advantage of reducing the amount of data required while shortening training time and improving performance when compared with models built from scratch.[26]

Convolutional neural networks require a fixed input size, whereas audio instances in our dataset were of variable length. To deal with this issue, Zero-padding was used to set the duration of each audio file to 50 s (the maximum length in our database). To raise the amount of information fed to the classifiers, type 1 (text reading) and type 2 ((a) phonation) audios were concatenated and used as a single input to the learning models.

All the audio recordings were first resampled to 8 kHz and then converted to Mel-spectrograms using the Librosa library in Python. The hop-length was 2048 samples, and the number of Mel coefficients was set to 196. The Mel spectrograms were passed through VGG19 convolutional neural network architecture provided by Keras, which was pretrained on the ImageNet database.[27] This approach, presented in figure 2, may be considered as a feature extraction step, as it converts audio recordings to 512 feature maps, each of a size 6×6, leading to a total of 18 432 features.

This large number of features is computationally expensive. Principal component analysis (PCA)[28] is therefore used for dimensionality reduction and to select the number of relevant components explaining the maximum of the variance in the data.

### Statistical analysis

We divided our data into 'fatigue' and 'no fatigue' groups based on the participant's reported answers for the inclusion and daily fatigue assessment of Predi-COVID. To characterise participants, descriptive statistics were used, which included means, SD for quantitative variables, and counts and percentages for qualitative variables. The two population groups (3gp (Android users) and m4a (iOS users)) were compared using a student test for continuous variables, and a $\chi^2$ test for categorical variables.

A 10-fold cross-validation procedure was conducted on the training cohort participants to evaluate four classification models (logistic regression, k-nearest neighbours, support vector machine (SVM) and soft voting classifier (VC), scikit-learn implementation in Python) at different regularisation levels via a grid search, with the following

**Table 2** Results of the prediction models

| Audio_format | Gender | ML model | Accuracy | Ov.precision | Precision_0 | Precision_1 | Ov.recall | Recall_0 | Recall_1 | Ov.f1score | f1-score_0 | f1-score_1 | Weighted AUC |
|---|---|---|---|---|---|---|---|---|---|---|---|---|---|
| 3gp (Android) | Female | LR | 0.77 | 0.77 | 0.81 | 0.73 | 0.77 | 0.76 | 0.77 | 0.77 | 0.78 | 0.75 | 0.85 |
| | | KNN | 0.72 | 0.73 | 0.7 | 0.77 | 0.72 | 0.87 | 0.55 | 0.72 | 0.78 | 0.64 | 0.76 |
| | | **SVM** | **0.8** | **0.8** | **0.8** | **0.79** | **0.8** | **0.84** | **0.74** | **0.8** | 0.82 | 0.77 | 0.86 |
| | | VC | 0.78 | 0.78 | 0.81 | 0.75 | 0.78 | 0.79 | 0.77 | 0.78 | 0.8 | 0.76 | 0.86 |
| | Male | LR | 0.78 | 0.79 | 0.87 | 0.5 | 0.78 | 0.85 | 0.53 | 0.79 | 0.86 | 0.52 | 0.81 |
| | | KNN | 0.83 | 0.83 | 0.83 | 0.8 | 0.83 | 0.98 | 0.27 | 0.79 | 0.9 | 0.4 | 0.84 |
| | | SVM | 0.84 | 0.83 | 0.88 | 0.67 | 0.84 | 0.93 | 0.53 | 0.83 | 0.9 | 0.59 | 0.82 |
| | | **VC** | **0.84** | **0.84** | **0.89** | **0.64** | **0.84** | **0.91** | **0.6** | **0.84** | 0.9 | 0.62 | 0.82 |
| m4a (iOS) | Female | LR | 0.72 | 0.72 | 0.8 | 0.56 | 0.72 | 0.77 | 0.61 | 0.72 | 0.79 | 0.58 | 0.75 |
| | | KNN | 0.68 | 0.65 | 0.72 | 0.5 | 0.68 | 0.86 | 0.29 | 0.65 | 0.78 | 0.37 | 0.67 |
| | | **SVM** | **0.79** | **0.79** | **0.81** | **0.75** | **0.79** | **0.91** | **0.55** | **0.79** | 0.86 | 0.64 | 0.79 |
| | | VC | 0.77 | 0.76 | 0.8 | 0.69 | 0.77 | 0.89 | 0.53 | 0.76 | 0.84 | 0.6 | 0.78 |
| | Male | LR | 0.73 | 0.74 | 0.83 | 0.48 | 0.73 | 0.8 | 0.54 | 0.73 | 0.81 | 0.51 | 0.8 |
| | | KNN | 0.89 | 0.89 | 0.89 | 0.89 | 0.89 | 0.97 | 0.65 | 0.88 | 0.93 | 0.76 | 0.81 |
| | | SVM | 0.85 | 0.84 | 0.86 | 0.76 | 0.85 | 0.95 | 0.58 | 0.84 | 0.9 | 0.67 | 0.85 |
| | | **VC** | **0.89** | **0.89** | **0.89** | **0.89** | **0.89** | **0.97** | **0.65** | **0.88** | 0.93 | 0.76 | 0.85 |

The selected models were selected using Recall_1 and weighted AUC and are highlighted in bold. Class 0: no fatigue, class 1: fatigue.
AUC, area under the curve; KNN, K-Nearest Neighbuors; LR, logistic regression; Ov, Overall; SVM, support vector machine; VC, voting classifier.

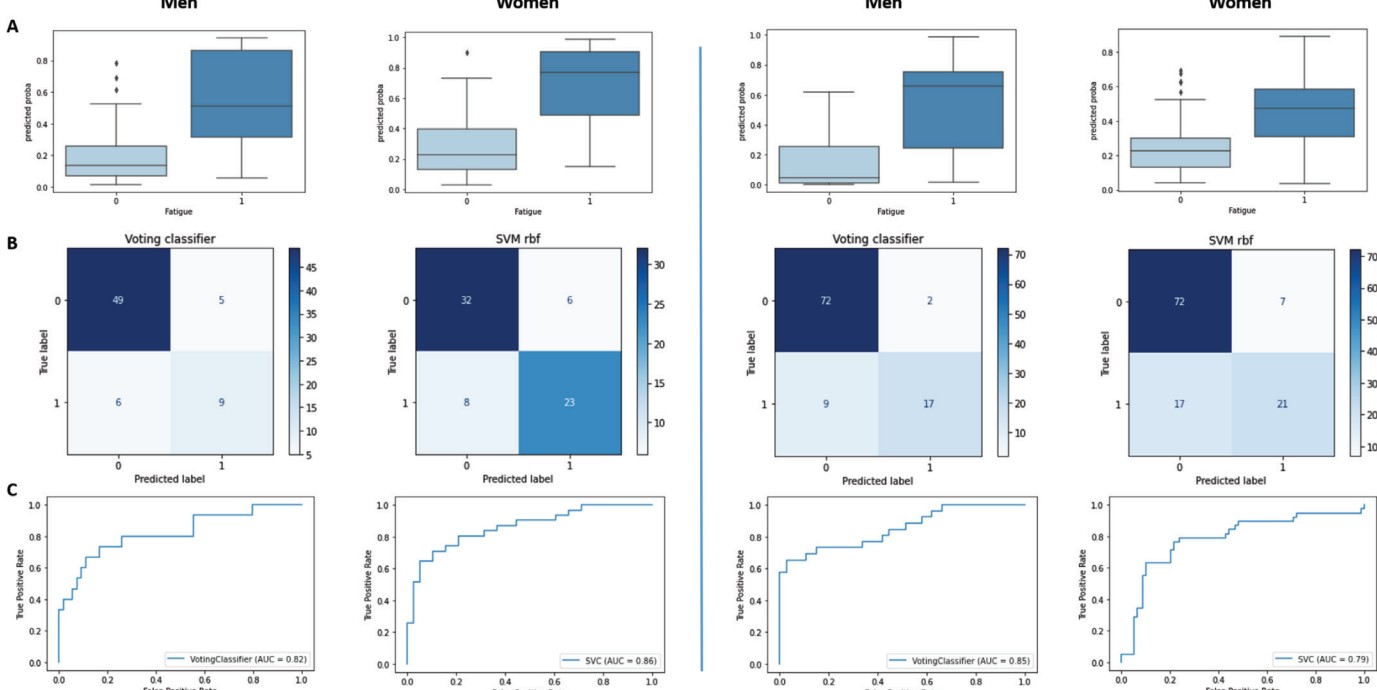

**Figure 3** Derivation of the digital fatigue vocal biomarker for android and iOS users.

evaluation metrics: area under the curve (AUC), accuracy, F1-score, precision and recall. The Brier score was also used to evaluate the calibration of the selected models.

The predicted probability of being classified as fatigued from the best model was considered as our final vocal biomarker, which may be used as a quantitative metric to monitor fatigue.

## RESULTS
### Study population characteristics
The final study population is composed of 296 participants of whom 165 were women (56%), with an average age of 40 years (SD=13). To record both audio types, 109 (37%) participants used Android smartphones (3gp format), whereas 187 (63%) used iOS devices (m4a format). We found no difference in the distribution of age, gender, body mass index, smoking, antibiotic usage and asthma, between the two types of devices (p>0.05). The overall rate of comorbidities in this study was relatively low: there were 31 (10%) participants who used antibiotics and only 12 (4%) participants with asthma. More details are shown in table 1.

Participants reported their fatigue status on average 6 days during the first 14 days of follow-up, resulting in the analysis of 1772 audio recordings for each audio type (type 1 and type 2) when all inclusion criteria were met, including 550 audio recordings for participants with fatigue. In both audio sets, women reported experiencing fatigue at a higher rate than men (p<0.001). Women constituted 155 (60%) of all fatigued Android users and 190 (67%) of all fatigued iOS users.

### Prediction models
We reduced the extracted features from Mel-spectrograms to 250 top components with PCA, explaining 97% and 99% of the variance in the data for iOS and Android audio sets, respectively. We then compared the performances of the machine learning algorithms to select the best models for the derivation of the vocal biomarkers.

The VC was the best model selected for the development of the vocal biomarker for male iOS users, with an AUC of 85% and overall accuracy, precision, recall and f1-score of 89%. The model selected for female iOS users was SVM with an overall precision of 79% and an AUC of 79%. For male Android users, the selected model is the VC with precision, recall an f1-score of 84%, and a weighted AUC of 82%. For female Android users, the SVM was selected with an overall precision of 80% and an AUC of 86%. More details are shown in table 2.

As shown in figure 3, the calibrations of the selected models were good (Mean Brier Scores=0.15, 0.12, 0.17 and 0.12, respectively, for Android female users, Android male users, iOS female users and iOS male users).

### Derivation of the digital fatigue vocal biomarker
Based on the model selected for each audio set, we derived the trained vocal biomarkers which quantitatively represent the probability of being labelled as fatigued.

## DISCUSSION
In this study, we built an AI-based pipeline to develop a vocal biomarker for both genders and both types of smartphones (male/female, Android/iOS) that effectively

recognise fatigued and non-fatigued participants with COVID-19.

We stratified the data to prevent data heterogeneity, which is considered contamination and makes it difficult to build a reliable and consistent classification model(s), resulting in poorer prediction performance. This contamination is caused by two factors: first, significant gender differences in fatigability, since it has previously been shown that men and women experience and report fatigue differently, and second, different microphone types incorporated in both smartphone devices used by the participants (iOS and Android), which have a direct impact on the quality of the recorded audios (machine learning algorithms separate the audio formats rather than the fatigue status if there is no constant microphone (see online supplemental material 2 for more details).

With the increased interest in remote voice analysis as a noninvasive and powerful telemedicine tool, various studies have been carried out, mostly in neurological disorders (eg, Parkinson's disease[11] and Alzheimer's disease[29] and mental health (eg, stress and depression)).[30] Recently, a significant research effort has evolved to employ respiratory sounds for COVID-19 and the main focus was on the use of cough[17 31] and breathing[32] to develop a COVID-19 screening tool. However, no previous work has been devoted to investigating the association of voice with COVID-19 symptoms, precisely fatigue.

Fatigue is one of the commonly reported symptoms of COVID-19 and Long-COVID syndrome,[33] which can persist regardless of how severe COVID-19's acute stage is.[34]

A variety of cerebral, peripheral and psychosocial factors[7 35] play a role in the development of fatigue. It may also occur from chronic inflammation in the brain and at neuromuscular junctions. New evidence shows that patients with Long-COVID syndrome continue to have higher measures of blood clotting, thrombosis,[36] which may also explain the persistence of fatigue. COVID-19 is associated with variations in airway resistance.[37] This narrowing of the airway is manifested in the increase in audible turbulence in both sighing and yawning, which is frequently associated with fatigue.[38]

Human voice is produced by the flow of air from the lungs through the larynx, which causes the vocal fold vibrations, generating a pulsating airstream.[39] The process is controlled by the laryngeal muscle activation[40] but involves the entire respiratory system to provide the air pressure necessary for phonation. Decreased pulmonary function in COVID-19 patients can cause reduced glottal airflow that is essential for normal voice production.[41] Furthermore, in case of increased fatigue, the voice production process may be additionally disturbed due to reduced laryngeal muscle tension, resulting in dysphonia that appears in up to 49% of COVID-19 patients.[41]

## Study limitations

This study has several limitations. First, although our data were stratified based on gender and smartphone devices, the mix of languages might also result in different voice features subsequently, in different model performances. There is presently no comparable dataset with similar audio recordings for further external validation of our findings. Thus, more data should be collected to improve the transferability of our vocal biomarker to other populations. Second, our data labelling was only based on a qualitative self-reported fatigue status. A fatigue severity scale would allow a quantitative assessment of fatigue severity in a uniform and unbiased way throughout all participants. Finally, time series voice analysis for each participant was not included in the study. More investigation, including time series analysis, would establish a personalised baseline for each participant, potentially enhancing the performance of our vocal biomarkers.

## CONCLUSION

In this study, we demonstrated the association between fatigue and voice in people with COVID-19 and developed a fatigue vocal biomarker that can accurately predict the presence of fatigue. These findings suggest that vocal biomarkers, digitally incorporated into telemonitoring technologies, might be used to identify and remotely monitor this symptom in patients suffering from COVID-19 as well as other chronic diseases.

**Acknowledgements** We thank all participants that accepted to be involved in the study, members that collaborated to the launch and monitoring of the Predi-COVID cohort, as well as its scientific committee, the IT team responsible for the development of the application, and the nurses in charge of recruitment, data collection and management on the field.

**Contributors** AB and GF had full access to all of the data in the study and took responsibility for the integrity of the data and the accuracy of the data analysis. GF is reponsible of the overall content as guarantor. GF, LZ and AF conceptualised and designed the study. AB, LZ, EH, AF, VD, PVN, GA and GF collected and analysed data and contributed to the interpretation. The statistical analysis was carried out by AB, LZ, EH and AF. AB drafted the initial manuscript. AB, LZ, EH, AF, VD, PVN, GA and GF critically revised the manuscript for more important intellectual content. GF obtained the funding. AF provided administrative, technical and material support. The corresponding author certifies that all listed authors fulfill the authorship criteria and that no other authors that meet the criteria have been omitted.

**Funding** The Predi-COVID study is supported by the Luxembourg National Research Fund (FNR) (Predi-COVID, grant number 14716273), the André Losch Foundation, and the Luxembourg Institute of Health.

**Competing interests** None declared.

**Patient and public involvement** Patients and/or the public were involved in the design, or conduct, or reporting, or dissemination plans of this research. Refer to the Methods section for further details.

**Patient consent for publication** Not applicable.

**Ethics approval** This study involves human participants and was approved by the National Research Ethics Committee of Luxembourg (study number 202003/07) provided ethics approval to the study in April 2020. Participants gave informed consent to participate in the study before taking part.

**Provenance and peer review** Not commissioned; externally peer reviewed.

**Data availability statement** Data are available in a public, open access repository. Audio data, datasets and source code used in this study are publicly available. Audio data available in Zenodo repository, (DOI: 10.5281/zenodo.5937844]Datasets and source code available in Github, (https://github.com/LIHVOICE/Predi_COVID_Fatigue_Vocal_Biomarker).

peer-reviewed. Any opinions or recommendations discussed are solely those of the author(s) and are not endorsed by BMJ. BMJ disclaims all liability and responsibility arising from any reliance placed on the content. Where the content includes any translated material, BMJ does not warrant the accuracy and reliability of the translations (including but not limited to local regulations, clinical guidelines, terminology, drug names and drug dosages), and is not responsible for any error and/or omissions arising from translation and adaptation or otherwise.

**ORCID iDs**
Abir Elbéji http://orcid.org/0000-0002-5293-4337
Gloria Aguayo http://orcid.org/0000-0002-5625-1664
Guy Fagherazzi http://orcid.org/0000-0001-5033-5966

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
