## [Reviewer comments · BMJ Open]

ARTICLE DETAILS

TITLE (PROVISIONAL)	Vocal biomarker predicts fatigue in people with COVID-19: results from the prospective Predi-COVID cohort study
AUTHORS	Elbéji, Abir; Zhang, Lu; Higa, Eduardo; Fischer, Aurélie; Despotovic, Vladimir; Nazarov, Petr V.; Aguayo, Gloria; Fagherazzi, Guy

VERSION 1 – REVIEW

REVIEWER	Gao, Yixiang University of Missouri
REVIEW RETURNED	12-Apr-2022

GENERAL COMMENTS	This is a interesting study that uses audio signals to help classify fatigue symptoms caused by Covid-19. It could have a great impact when applied in a more general level. However, a few questions that I would like the author to clarify: 1. In "Audio pre-processing", when the author mentioned about excluding poor quality audios, were there a more detailed criteria? For example, the audio was too noisy, too much silence, too loud etc. It was a very vague description.2. Why zero padding instead of using the shortest signal length as the standard? In some sense, the zero pads could also be recognized as features for the particular signal.3. In "feature extraction" section, the phrase "type 1 and type 2 audios" were being referenced and that was the only time it appeared in the paper which cause some confusion of what exactly those audios are, please provide more claffications.
--

REVIEWER	Berardi, Mark University of Bonn
REVIEW RETURNED	10-May-2022

GENERAL COMMENTS	This manuscript presents the development of a vocal biomarker for fatigue in individuals with COVID-19. This is a valued work as fatigue is a significant symptom that affects those with COVID-19 with potential application to long-term effects of COVID-19. The authors present an approach to classifying the probability of fatigue based on voice recordings. The manuscript provides an important contribution to mobile-based voice analysis for symptom detection. The greatest strengths of the manuscript are the well-defined objective and methods. The objective is clear and specific, and the approach
---

	has sufficient detail to be repeatable. The major weakness of the paper is in the statistical application. Here a Student's t-test is used to compare the means of the distribution of the derived vocal biomarker between non-fatigued and fatigued samples within a stratified group. However, this statistical test must meet certain assumptions and it is unclear that these assumptions are met. Specifically, the assumptions of independence, normality, and homogeneity of variances need to be reported as not being violated for this test. From my understanding of the methods, at least one of these assumptions is likely violated. For example, the manuscript on page 7 lines 14-16 state "As a result, several audio recordings for a single participant were available for both audio types," which means the samples are likely not independent. An appropriate statistical test should be used. Additionally, please report the sample size for each statistical test and report the effect sizes and confidence intervals of the tests. This will inform whether there is a meaningful difference in the means of the groups and is best practice for reporting statistical tests. If the current results are confirmed with an appropriate statistical model, then the manuscript will be a valued contribution to the development and application of voice-based mobile screenings.
--	--

VERSION 1 – AUTHOR RESPONSE

Reviewer: 1

This is an interesting study that uses audio signals to help classify fatigue symptoms caused by Covid-19. It could have a great impact when applied in a more general level.

Response: We thank Reviewer #1 for this positive feedback.

However, a few questions that I would like the author to clarify:

1. In "Audio pre-processing", when the author mentioned about excluding poor quality audios, were there a more detailed criteria? For example, the audio was too noisy, too much silence, too loud etc. It was a very vague description.

Response: Thank you for giving us the opportunity to clarify this point. We have manually checked the outliers from the DBSCAN clustering and have excluded the audios with either 1) too much noise, 2) incorrect text reading, or 3) type 1 and type 2 audios mixed or 4) extended silence in the middle. We have now added this clarification in "Audio pre-processing": "Then, an audio clustering (DBSCAN) on basic features (duration, average, sum, and standard deviation of signal power, and fundamental frequency) was performed to detect outliers that were manually checked while excluding poor quality audios with 1) too noisy, 2) incorrect text reading, 3) type 1 and type 2 audios mixed, or 4) extended silence in the middle."

2. Why zero padding instead of using the shortest signal length as the standard? In some sense, the zero pads could also be recognized as features for the particular signal.

Response: Thank you for bringing this up. Previous works in speech signal processing have shown that the loss of information from truncation influences performance more negatively than the addition

of information via zero-padding [Yoon, S.-H.; Yu, H.-J. *A Simple Distortion-Free Method to Handle Variable Length Sequences for Recurrent Neural Networks in Text-Dependent Speaker Verification. Appl. Sci.* 2020, 10, 4092.]. This was indeed confirmed in our case, since adopting the shortest signal length resulted in unstable results across all groups due to considerable signal loss (weighted AUCs were below 70%). Padding with silence allows us to evaluate the entirety of audio recordings and use all available information. Since we use a convolutional neural network for feature extraction that identifies patterns locally around the kernel, the zero padding should not have a significant impact on the model performance. On the other hand, it is true that zero-padding adds additional information to the signal that still requires processing, therefore making it less computationally efficient. The analysis of different signal padding/truncation approaches to performance and computational complexity of the models was, however, out of the scope of this paper.

3. In "feature extraction" section, the phrase "type 1 and type 2 audios" were being referenced and that was the only time it appeared in the paper which cause some confusion of what exactly those audios are, please provide more clarifications.

Response: Thank you for pointing this out. Both audio types were explicitly defined in the section "Data collection" previous to "Feature extraction". But to avoid confusion, we have now rephrased the initial sentence as such: "type 1 (text reading) and type 2 ([a] phonation) audios were concatenated and used as a single input to the learning models."

Reviewer 2:

This manuscript presents the development of a vocal biomarker for fatigue in individuals with COVID-19. This is a valued work as fatigue is a significant symptom that affects those with COVID-19 with potential application to long-term effects of COVID-19. The authors present an approach to classifying the probability of fatigue based on voice recordings.

The manuscript provides an important contribution to mobile-based voice analysis for symptom detection. The greatest strengths of the manuscript are the well-defined objective and methods. The objective is clear and specific, and the approach has sufficient detail to be repeatable.

Response: We thank Reviewer #2 for this positive feedback

Here a Student's t-test is used to compare the means of the distribution of the derived vocal biomarker between non-fatigued and fatigued samples within a stratified group. However, this statistical test must meet certain assumptions and it is unclear that these assumptions are met. Specifically, the assumptions of independence, normality, and homogeneity of variances need to be reported as not being violated for this test. From my understanding of the methods, at least one of these assumptions is likely violated. For example, the manuscript on page 7 lines 14-16 state "As a result, several audio recordings for a single

participant were available for both audio types," which means the samples are likely not independent. An appropriate statistical test should be used. Additionally, please report the sample size for each statistical test and report the effect sizes and confidence intervals of the tests. This will inform whether there is a meaningful difference in the means of the groups and is best practice for reporting statistical tests.

Response: Thank you for raising this point. We fully agree with Reviewer #2; the independence assumption was violated. It was virtually impossible for us to have independent groups since Predi-COVID is a prospective cohort study where symptoms such as fatigue evolve over time and, as such, a given study participant can theoretically contribute to both groups (fatigue/non-fatigue). To avoid

further confusion, we prefer to remove the p-value and the corresponding statistical test. We are convinced that the metrics mentioned in the paper (Accuracy, precision, recall, F1-score, and weighted AUC) are sufficient to assess our models in distinguishing between COVID-19 participants with fatigue and those without fatigue.

VERSION 2 – REVIEW

REVIEWER	Berardi, Mark University of Bonn
REVIEW RETURNED	22-Sep-2022

GENERAL COMMENTS	In the response to reviewers, it is stated that "We are convinced that the metrics mentioned in the paper (Accuracy, precision, recall, F1-score, and weighted AUC) are sufficient to assess our models in distinguishing between COVID-19 participants with fatigue and those without fatigue." I agree with this, however, this needs to be more clear in the manuscript! Currently, it reads as the quality of the vocal biomarker is qualified by a t-test but the results of the t-test are omitted. A statement similar to this response needs to be included to provide justification for the conclusion given.
--

VERSION 2 – AUTHOR RESPONSE

Reviewer: 2

In the response to reviewers, it is stated that "We are convinced that the metrics mentioned in the paper (Accuracy, precision, recall, F1-score, and weighted AUC) are sufficient to assess our models in distinguishing between COVID-19 participants with fatigue and those without fatigue." I agree with this, however, this needs to be more clear in the manuscript! Currently, it reads as the quality of the vocal biomarker is qualified by a t-test but the results of the t-test are omitted. A statement similar to this response needs to be included to provide justification for the conclusion given.

Response: We thank Reviewer #2. We have now categorically stated on the paper that we rely on these metrics (Accuracy, precision, recall, F1-score, and weighted AUC as well as Brier score) to assess the quality of the vocal biomarker.